# Competitive Gradient Descent

**Florian Schäfer**
Computing and Mathematical Sciences
California Institute of Technology
Pasadena, CA 91125
florian.schaefer@caltech.edu

**Anima Anandkumar**
Computing and Mathematical Sciences
California Institute of Technology
Pasadena, CA 91125
anima@caltech.edu

## Abstract

We introduce a new algorithm for the numerical computation of Nash equilibria of competitive two-player games. Our method is a natural generalization of gradient descent to the two-player setting where the update is given by the Nash equilibrium of a regularized bilinear local approximation of the underlying game. It avoids oscillatory and divergent behaviors seen in alternating gradient descent. Using numerical experiments and rigorous analysis, we provide a detailed comparison to methods based on *optimism* and *consensus* and show that our method avoids making any unnecessary changes to the gradient dynamics while achieving exponential (local) convergence for (locally) convex-concave zero sum games. Convergence and stability properties of our method are robust to strong interactions between the players, without adapting the stepsize, which is not the case with previous methods. In our numerical experiments on non-convex-concave problems, existing methods are prone to divergence and instability due to their sensitivity to interactions among the players, whereas we never observe divergence of our algorithm. The ability to choose larger stepsizes furthermore allows our algorithm to achieve faster convergence, as measured by the number of model evaluations.

## 1 Introduction

**Competitive optimization:** Whereas traditional optimization is concerned with a single agent trying to optimize a cost function, competitive optimization extends this problem to the setting of multiple agents each trying to minimize their own cost function, which in general depends on the actions of all agents. The present work deals with the case of two such agents:

$$\min_{x \in \mathbb{R}^m} f(x, y), \quad \min_{y \in \mathbb{R}^n} g(x, y) \tag{1}$$

for two functions $f, g : \mathbb{R}^m \times \mathbb{R}^n \longrightarrow \mathbb{R}$.
In single agent optimization, the solution of the problem consists of the minimizer of the cost function. In competitive optimization, the right definition of *solution* is less obvious, but often one is interested in computing Nash– or strategic equilibria: Pairs of strategies, such that no player can decrease their costs by unilaterally changing their strategies. If $f$ and $g$ are not convex, finding a global Nash equilibrium is typically impossible and instead we hope to find a "good" local Nash equilibrium.

**The benefits of competition:** While competitive optimization problems arise naturally in mathematical economics and game/decision theory (Nisan et al., 2007), they also provide a highly expressive and transparent language to formulate algorithms in a wide range of domains. In optimization (Bertsimas et al., 2011) and statistics (Huber and Ronchetti, 2009) it has long been observed that competitive optimization is a natural way to encode robustness requirements of algorithms. More recently, researchers in machine learning have been using multi-agent optimization to design highly flexible objective functions for reinforcement learning (Liu et al., 2016; Pfau and Vinyals, 2016;

Pathak et al., 2017; Wayne and Abbott, 2014; Vezhnevets et al., 2017) and generative models (Goodfellow et al., 2014). We believe that this approach has still a lot of untapped potential, but its full realization depends crucially on the development of efficient and reliable algorithms for the numerical solution of competitive optimization problems.

**Gradient descent/ascent and the cycling problem:** For differentiable objective functions, the most naive approach to solving (1) is gradient descent ascent (GDA), whereby both players independently change their strategy in the direction of steepest descent of their cost function. Unfortunately, this procedure features oscillatory or divergent behavior even in the simple case of a bilinear game $(f(x, y) = x^\top y = -g(x, y))$ (see Figure 2). In game-theoretic terms, GDA lets both players choose their new strategy optimally with respect to the last move of the other player. Thus, the cycling behaviour of GDA is not surprising: It is the analogue of *"Rock! Paper! Scissors! Rock! Paper! Scissors! Rock! Paper!..."* in the eponymous hand game. While gradient descent is a reliable basic *workhorse* for single-agent optimization, GDA can not play the same role for competitive optimization. At the moment, the lack of such a *workhorse* greatly hinders the broader adoption of methods based on competition.

**Existing works:** Most existing approaches to stabilizing GDA follow one of three lines of attack.
In the special case $f = -g$, the problem can be written as a minimization problem $\min_x F(x)$, where $F(x) := \max_y f(x, y)$. For certain structured problems, Gilpin et al. (2007) use techniques from convex optimization (Nesterov, 2005) to minimize the implicitly defined $F$. For general problems, the two-scale update rules proposed in Goodfellow et al. (2014); Heusel et al. (2017); Metz et al. (2016) can be seen as an attempt to approximate $F$ and its gradients.
In GDA, players pick their next strategy based on the last strategy picked by the other players. Methods based on *follow the regularized leader* (Shalev-Shwartz and Singer, 2007; Grnarova et al., 2017), *fictitious play* (Brown, 1951), *predictive updates* (Yadav et al., 2017), *opponent learning awareness* (Foerster et al., 2018), and *optimism* (Rakhlin and Sridharan, 2013; Daskalakis et al., 2017; Mertikopoulos et al., 2019) propose more sophisticated heuristics that the players could use to predict each other's next move. Algorithmically, many of these methods can be considered variations of the *extragradient method* (Korpelevich, 1977)(see also Facchinei and Pang (2003)[Chapter 12]). Finally, some methods directly modify the gradient dynamics, either by promoting convergence through gradient penalties (Mescheder et al., 2017), or by attempting to disentangle convergent *potential* parts from rotational *Hamiltonian* parts of the vector field (Balduzzi et al., 2018; Letcher et al., 2019; Gemp and Mahadevan, 2018).

**Our contributions:** Our main *conceptual* objection to most existing methods is that they lack a clear game-theoretic motivation, but instead rely on the ad-hoc introduction of additional assumptions, modifications, and model parameters.
Their main *practical* shortcoming is that to avoid divergence the stepsize has to be chosen inversely proportional to the magnitude of the interaction of the two players (as measured by $D^2_{xy}f$, $D^2_{xy}g$). On the one hand, the small stepsize results in slow convergence. On the other hand, a stepsize small enough to prevent divergence will not be known in advance in most problems. Instead it has to be discovered through tedious trial and error, which is further aggravated by the lack of a good diagnostic for improvement in multi-agent optimization (which is given by the objective function in single agent optimization).
We alleviate the above mentioned problems by introducing a novel algorithm, *competitive gradient descent* (CGD) that is obtained as a natural extension of gradient descent to the competitive setting. Recall that in the single player setting, the gradient descent update is obtained as the optimal solution to a regularized linear approximation of the cost function. In the same spirit, the update of CGD is given by the Nash equilibrium of a regularized *bilinear* approximation of the underlying game. The use of a bilinear– as opposed to linear approximation lets the local approximation preserve the competitive nature of the problem, significantly improving stability. We prove (local) convergence results of this algorithm in the case of (locally) convex-concave zero-sum games. We also show that stronger interactions between the two players only improve convergence, without requiring an adaptation of the stepsize. In comparison, the existing methods need to reduce the stepsize to match the increase of the interactions to avoid divergence, which we illustrate on a series of polynomial test cases considered in previous works.
We begin our numerical experiments by trying to use a GAN on a bimodal Gaussian mixture model. Even in this simple example, trying five different (constant) stepsizes under RMSProp, the existing

methods diverge. The typical solution would be to decay the learning rate. However even with a constant learning rate, CGD succeeds with all these stepsize choices to approximate the main features of the target distribution. In fact, throughout our experiments we *never* saw CGD diverge. In order to measure the convergence speed more quantitatively, we next consider a nonconvex matrix estimation problem, measuring computational complexity in terms of the number of gradient computations performed. We observe that all methods show improved speed of convergence for larger stepsizes, with CGD roughly matching the convergence speed of optimistic gradient descent (Daskalakis et al., 2017), at the same stepsize. However, as we increase the stepsize, other methods quickly start diverging, whereas CGD continues to improve, thus being able to attain significantly better convergence rates (more than two times as fast as the other methods in the noiseless case, with the ratio increasing for larger and more difficult problems). For small stepsize or games with weak interactions on the other hand, CGD automatically invests less computational time per update, thus gracefully transitioning to a cheap correction to GDA, at minimal computational overhead. We believe that the robustness of CGD makes it an excellent candidate for the fast and simple training of machine learning systems based on competition, hopefully helping them reach the same level of automatization and ease-of-use that is already standard in minimization based machine learning.

## 2  Competitive gradient descent

We propose a novel algorithm, which we call *competitive gradient descent* (CGD), for the solution of competitive optimization problems $\min_{x \in \mathbb{R}^m} f(x, y)$, $\min_{y \in \mathbb{R}^n} g(x, y)$, where we have access to function evaluations, gradients, and Hessian-vector products of the objective functions. [1]

---
**Algorithm 1:** Competitive Gradient Descent (CGD)

**for** $0 \leq k \leq N - 1$ **do**

> $x_{k+1} = x_k - \eta \left( \mathrm{Id} - \eta^2 D_{xy}^2 f D_{yx}^2 g \right)^{-1} \left( \nabla_x f - \eta D_{xy}^2 f \nabla_y g \right);$
> $y_{k+1} = y_k - \eta \left( \mathrm{Id} - \eta^2 D_{yx}^2 g D_{xy}^2 f \right)^{-1} \left( \nabla_y g - \eta D_{yx}^2 g \nabla_x f \right);$

**return** $(x_N, y_N);$

---

**How to linearize a game:** To motivate this algorithm, we remind ourselves that gradient descent with stepsize $\eta$ applied to the function $f : \mathbb{R}^m \longrightarrow \mathbb{R}$ can be written as

$$x_{k+1} = \mathrm{argmin}_{x \in \mathbb{R}^m} (x^\top - x_k^\top) \nabla_x f(x_k) + \frac{1}{2\eta} \|x - x_k\|^2.$$

This models a (single) player solving a local linear approximation of the (minimization) game, subject to a quadratic penalty that expresses her limited confidence in the global accuracy of the model. The natural generalization of this idea to the competitive case should then be given by the two players solving a local approximation of the true game, both subject to a quadratic penalty that expresses their limited confidence in the accuracy of the local approximation.

In order to implement this idea, we need to find the appropriate way to generalize the linear approximation in the single agent setting to the competitive setting: *How to linearize a game?*.

**Linear or Multilinear:** GDA answers the above question by choosing a linear approximation of $f, g : \mathbb{R}^m \times \mathbb{R}^n \longrightarrow \mathbb{R}$. This seemingly natural choice has the flaw that linear functions can not express any interaction between the two players and are thus unable to capture the competitive nature of the underlying problem. From this point of view it is not surprising that the convergent modifications of GDA are, implicitly or explicitly, based on higher order approximations (see also (Li et al., 2017)). An equally valid generalization of the linear approximation in the single player setting is to use a *bilinear* approximation in the two-player setting. Since the bilinear approximation is the lowest order approximation that can capture some interaction between the two players, we argue that the natural generalization of gradient descent to competitive optimization is not GDA, but rather the

update rule $(x_{k+1}, y_{k+1}) = (x_k, y_k) + (x, y)$, where $(x, y)$ is a Nash equilibrium of the game [2]

$$\min_{x \in \mathbb{R}^m} x^\top \nabla_x f + x^\top D_{xy}^2 f y + y^\top \nabla_y f + \frac{1}{2\eta} x^\top x$$

$$\min_{y \in \mathbb{R}^n} y^\top \nabla_y g + y^\top D_{yx}^2 g x + x^\top \nabla_x g + \frac{1}{2\eta} y^\top y. \tag{2}$$

Indeed, the (unique) Nash equilibrium of the Game (2) can be computed in closed form.

**Theorem 2.1.** *Among all (possibly randomized) strategies with finite first moment, the only Nash equilibrium of the Game* (2) *is given by*

$$x = -\eta \left( \mathrm{Id} - \eta^2 D_{xy}^2 f D_{yx}^2 g \right)^{-1} \left( \nabla_x f - \eta D_{xy}^2 f \nabla_y g \right) \tag{3}$$

$$y = -\eta \left( \mathrm{Id} - \eta^2 D_{yx}^2 g D_{xy}^2 f \right)^{-1} \left( \nabla_y g - \eta D_{yx}^2 g \nabla_x f \right),$$

*given that the matrix inverses in the above expression exist.* [3]

*Proof.* Let $X, Y$ be randomized strategies. By subtracting and adding $\mathbb{E}[X]^2/(2\eta), \mathbb{E}[Y]^2/(2\eta)$, and taking expectations, we can rewrite the game as

$$\min_{\mathbb{E}[X] \in \mathbb{R}^m} \mathbb{E}[X]^\top \nabla_x f + \mathbb{E}[X]^\top D_{xy}^2 f \mathbb{E}[Y] + \mathbb{E}[Y]^\top \nabla_y f + \frac{1}{2\eta} \mathbb{E}[X]^\top \mathbb{E}[X] + \frac{1}{2\eta} \mathrm{Var}[X]$$

$$\min_{\mathbb{E}[Y] \in \mathbb{R}^n} \mathbb{E}[Y]^\top \nabla_y g + \mathbb{E}[Y]^\top D_{yx}^2 g \mathbb{E}[X] + \mathbb{E}[X]^\top \nabla_x g + \frac{1}{2\eta} \mathbb{E}[Y]^\top \mathbb{E}[Y] + \frac{1}{2\eta} \mathrm{Var}[Y].$$

Thus, the objective value for both players can always be improved by decreasing the variance while keeping the expectation the same, meaning that the optimal value will always (and only) be achieved by a deterministic strategy. We can then replace the $\mathbb{E}[X], \mathbb{E}[Y]$ with $x, y$, set the derivative of the first expression with respect to $x$ and of the second expression with respect to $y$ to zero, and solve the resulting system of two equations for the Nash equilibrium $(x, y)$. $\qquad \square$

According to Theorem 2.1, the Game (2) has exactly one optimal pair of strategies, which is deterministic. Thus, we can use these strategies as an update rule, generalizing the idea of local optimality from the single– to the multi agent setting and obtaining Algorithm 1.

**What I think that they think that I think ... that they do**: Another game-theoretic interpretation of CGD follows from the observation that its update rule can be written as

$$\begin{pmatrix} \Delta x \\ \Delta y \end{pmatrix} = - \begin{pmatrix} \mathrm{Id} & \eta D_{xy}^2 f \\ \eta D_{yx}^2 g & \mathrm{Id} \end{pmatrix}^{-1} \begin{pmatrix} \nabla_x f \\ \nabla_y g \end{pmatrix}. \tag{4}$$

Applying the expansion $\lambda_{\max}(A) < 1 \Rightarrow (\mathrm{Id} - A)^{-1} = \lim_{N \to \infty} \sum_{k=0}^N A^k$ to the above equation, we observe that the first partial sum $(N = 0)$ corresponds to the optimal strategy if the other player's strategy stays constant (GDA). The second partial sum $(N = 1)$ corresponds to the optimal strategy if the other player thinks that the other player's strategy stays constant (LCGD, see Figure 1). The third partial sum $(N = 2)$ corresponds to the optimal strategy if the other player thinks that the other player thinks that the other player's strategy stays constant, and so forth, until the Nash equilibrium is recovered in the limit. For small enough $\eta$, we could use the above series expansion to solve for $(\Delta x, \Delta y)$, which is known as Richardson iteration and would recover high order LOLA (Foerster et al., 2018). However, expressing it as a matrix inverse will allow us to use optimal Krylov subspace methods to obtain far more accurate solutions with fewer gradient evaluations.

**Rigorous results on convergence and local stability:** We will now show some basic convergence results for CGD, the proofs of which we defer to the appendix. Our results are restricted to the case of a zero-sum game $(f = -g)$, but we expect that they can be extended to games that are dominated by competition. To simplify notation, we define

$$\bar{D} := (\mathrm{Id} + \eta^2 D_{xy}^2 f D_{yx}^2 f)^{-1} \eta^2 D_{xy}^2 f D_{yx}^2 f, \quad \tilde{D} := (\mathrm{Id} + \eta^2 D_{yx}^2 f D_{xy}^2 f)^{-1} \eta^2 D_{yx}^2 f D_{xy}^2 f.$$

We furthermore define the spectral function $h_\pm(\lambda) := \min(3\lambda, \lambda)/2$.

**Theorem 2.2.** *If $f$ is two times continiously differentiable with L-Lipschitz continuous mixed Hessian, $f$ is convex-concave or $D^2_{xx}f, D^2_{yy}f$ are L-Lipschitz continuous, and the diagonal blocks of its Hessian are bounded as $\eta\|D^2_{xx}f\|, \eta\|D^2_{yy}f\| \leq 1$, we have*

$$\|\nabla_x f(x_{k+1}, y_{k+1})\|^2 + \|\nabla_y f(x_{k+1}, y_{k+1})\|^2 - \|\nabla_x f\|^2 - \|\nabla_y f\|^2 \leq$$
$$- \nabla_x f^\top \left(\eta h_\pm \left(D^2_{xx}f\right) + \bar{D} - 32L\eta^2\|\nabla_x f\|\right)\nabla_x f - \nabla_y f^\top \left(\eta h_\pm \left(-D^2_{yy}f\right) + \tilde{D} - 32L\eta^2\|\nabla_y f\|\right)\nabla_y f$$

Under suitable assumptions on the curvature of $f$, Theorem 2.2 implies results on the convergence of CGD.

**Corollary 2.2.1.** *Under the assumptions of Theorem 2.2, if for $\alpha > 0$*

$$\left(\eta h_\pm \left(D^2_{xx}f\right) + \bar{D} - 32L\eta^2\|\nabla_x f(x_0, y_0)\|\right), \left(\eta h_\pm \left(-D^2_{yy}f\right) + \tilde{D} - 32L\eta^2\|\nabla_y f(x_0, y_0)\|\right) \succeq \alpha\,\mathrm{Id},$$

*for all $(x, y) \in \mathbb{R}^{m+n}$, then CGD started in $(x_0, y_0)$ converges at exponential rate with exponent $\alpha$ to a critical point.*

Furthermore, we can deduce the following local stability result.

**Theorem 2.3.** *Let $(x^*, y^*)$ be a critical point $((\nabla_x f, \nabla_y f) = (0, 0))$ and assume furthermore that $\lambda_{\min} := \min\left(\lambda_{\min}\left(\eta D^2_{xx}f + \bar{D}\right), \lambda_{\min}\left(-\eta D^2_{yy}f + \bar{D}\right)\right) > 0$ and $f \in C^2(\mathbb{R}^{m+n})$ with Lipschitz continuous mixed Hessian. Then there exists a neighbourhood $\mathcal{U}$ of $(x^*, y^*)$, such that CGD started in $(x_1, y_1) \in \mathcal{U}$ converges to a point in $\mathcal{U}$ at an exponential rate that depends only on $\lambda_{\min}$.*

The results on local stability for existing modifications of GDA, including those of (Mescheder et al., 2017; Daskalakis et al., 2017; Mertikopoulos et al., 2019) (see also Liang and Stokes (2018)) all require the stepsize to be chosen inversely proportional to an upper bound on $\sigma_{\max}(D^2_{xy}f)$ and indeed we will see in our experiments that the existing methods are prone to divergence under strong interactions between the two players (large $\sigma_{\max}(D^2_{xy}f)$). In contrast to these results, our convergence results *only improve* as the interaction between the players becomes stronger.

**Why not use $D^2_{xx}f$ and $D^2_{yy}g$?:** The use of a bilinear approximation that contains some, but not all second order terms is unusual and begs the question why we do not include the diagonal blocks of the Hessian in Equation (4) resulting in the damped and regularized Newton's method

$$\begin{pmatrix}\Delta x \\ \Delta y\end{pmatrix} = -\begin{pmatrix}\mathrm{Id} + \eta D^2_{xx}f & \eta D^2_{xy}f \\ \eta D^2_{yx}g & \mathrm{Id} + \eta D^2_{yy}g\end{pmatrix}^{-1}\begin{pmatrix}\nabla_x f \\ \nabla_y g\end{pmatrix}. \tag{5}$$

For the following reasons we believe that the bilinear approximation is preferable both from a practical and conceptual point of view.

• *Conditioning of matrix inverse:* One advantage of competitive gradient descent is that in many cases, including all zero-sum games, the condition number of the matrix inverse in Algorithm 1 is bounded above by $\eta^2\|D_{xy}\|^2$. If we include the diagonal blocks of the Hessian in a non-convex-concave problem, the matrix can even be singular as soon as $\eta\|D^2_{xx}f\| \geq 1$ or $\eta\|D^2_{yy}g\| \geq 1$.

• *Irrational updates:* We can only expect the update rule (5) to correspond to a local Nash equilibrium if the problem is convex-concave or $\eta\|D^2_{xx}f\|, \eta\|D^2_{yy}g\| < 1$. If these conditions are violated it can instead correspond to the players playing their *worst* as opposed to best strategy based on the quadratic approximation, leading to behavior that contradicts the game-interpretation of the problem.

• *Lack of regularity:* For the inclusion of the diagonal blocks of the Hessian to be helpful at all, we need to make additional assumptions on the regularity of $f$, for example by bounding the Lipschitz constants of $D^2_{xx}f$ and $D^2_{yy}g$. Otherwise, their value at a given point can be totally uninformative about the global structure of the loss functions (consider as an example the minimization of $x \mapsto x^2 + \epsilon^{3/2}\sin(x/\epsilon)$ for $\epsilon \ll 1$). Many problems in competitive optimization, including GANs, have the form $f(x, y) = \Phi(\mathcal{G}(x), \mathcal{D}(y)), g(x, y) = \Theta(\mathcal{G}(x), \mathcal{D}(y))$, where $\Phi, \Theta$ are *smooth* and *simple*, but $\mathcal{G}$ and $\mathcal{D}$ might only have first order regularity. In this setting, the bilinear approximation has the advantage of fully exploiting the first order information of $\mathcal{G}$ and $\mathcal{D}$, without assuming them to have higher order regularity. This is because the bilinear approximations of $f$ and $g$ then contains only the first derivatives of $\mathcal{G}$ and $\mathcal{D}$, while the quadratic approximation contains the second derivatives $D^2_{xx}\mathcal{G}$ and $D^2_{yy}\mathcal{D}$ and therefore needs stronger regularity assumptions on $\mathcal{G}$ and $\mathcal{D}$ to be effective.

$$
\begin{aligned}
\text{GDA:} \quad & \Delta x = & -\nabla_x f & & & \\
\text{LCGD:} \quad & \Delta x = & -\nabla_x f & \quad -\eta D_{xy}^2 f \nabla_y f & & \\
\text{SGA:} \quad & \Delta x = & -\nabla_x f & \quad -\gamma D_{xy}^2 f \nabla_y f & & \\
\text{ConOpt:} \quad & \Delta x = & -\nabla_x f & \quad -\gamma D_{xy}^2 f \nabla_y f & \quad -\gamma D_{xx}^2 f \nabla_x f & \\
\text{OGDA:} \quad & \Delta x \approx & -\nabla_x f & \quad -\eta D_{xy}^2 f \nabla_y f & \quad +\eta D_{xx}^2 f \nabla_x f & \\
\text{CGD:} \quad & \Delta x = \left(\text{Id} + \eta^2 D_{xy}^2 f D_{yx}^2 f\right)^{-1} & \left(-\nabla_x f \right. & \quad \left. -\eta D_{xy}^2 f \nabla_y f \right. & & \left. \right)
\end{aligned}
$$

Figure 1: The update rules of the first player for (from top to bottom) GDA, LCGD, ConOpt, OGDA, and CGD, in a zero-sum game ($f = -g$).

- *No spurious symmetry:* One reason to favor full Taylor approximations of a certain order in single-player optimization is that they are invariant under changes of the coordinate system. For competitive optimization, a change of coordinates of $(x, y) \in \mathbb{R}^{m+n}$ can correspond, for instance, to taking a decision variable of one player and giving it to the other player. This changes the underlying game significantly and thus we do *not* want our approximation to be invariant under this transformation. Instead, we want our local approximation to only be invariant to coordinate changes of $x \in \mathbb{R}^m$ and $y \in \mathbb{R}^n$ *in separation*, that is to block-diagonal coordinate changes on $\mathbb{R}^{m+n}$. *Mixed* order approximations (bilinear, biquadratic, etc.) have exactly this invariance property and thus are the natural approximation for two-player games.

While we are convinced that the right notion of first order competitive optimization is given by quadratically regularized bilinear approximations, we believe that the right notion of second order competitive optimization is given by *cubically* regularized *biquadratic* approximations, in the spirit of Nesterov and Polyak (2006).

## 3    Consensus, optimism, or competition?

We will now show that many of the convergent modifications of GDA correspond to different subsets of four common ingredients. *Consensus optimization* (ConOpt) (Mescheder et al., 2017), penalises the players for non-convergence by adding the squared norm of the gradient at the next location, $\gamma \| \nabla_x f(x_{k+1}, y_{k+1}), \nabla_x f(x_{k+1}, y_{k+1}) \|^2$ to both player's loss function (here $\gamma \geq 0$ is a hyperparameter). As we see in Figure 1, the resulting gradient field has two additional Hessian corrections. Balduzzi et al. (2018); Letcher et al. (2019) observe that any game can be written as the sum of a *potential game* (that is easily solved by GDA), and a *Hamiltonian game* (that is easily solved by ConOpt). Based on this insight, they propose *symplectic gradient adjustment* that applies (in its simplest form) ConOpt only using the skew-symmetric part of the Hessian, thus alleviating the problematic tendency of ConOpt to converge to spurious solutions. The same algorithm was independently discovered by Gemp and Mahadevan (2018), who also provide a detailed analysis in the case of linear-quadratic GANs.
Daskalakis et al. (2017) proposed to modify GDA as

$$
\begin{aligned}
\Delta x &= -\left(\nabla_x f(x_k, y_k) + \left(\nabla_x f(x_k, y_k) - \nabla_x f(x_{k-1}, y_{k-1})\right)\right) \\
\Delta y &= -\left(\nabla_y g(x_k, y_k) + \left(\nabla_y g(x_k, y_k) - \nabla_y g(x_{k-1}, y_{k-1})\right)\right),
\end{aligned}
$$

which we will refer to as optimistic gradient descent ascent (OGDA). By interpreting the differences appearing in the update rule as finite difference approximations to Hessian vector products, we see that (to leading order) OGDA corresponds to yet another second order correction of GDA (see Figure 1). It will also be instructive to compare the algorithms to linearized competitive gradient descent (LCGD), which is obtained by skipping the matrix inverse in CGD (which corresponds to taking only the leading order term in the limit $\eta D_{xy}^2 f \to 0$) and also coincides with first order LOLA (Foerster et al., 2018). As illustrated in Figure 1, these six algorithms amount to different subsets of the following four terms.

1. The *gradient term* $-\nabla_x f$, $\nabla_y f$ which corresponds to the most immediate way in which the players can improve their cost.

2. The *competitive term* $-D_{xy}f\nabla_y f$, $D_{yx}f\nabla_x f$ which can be interpreted either as anticipating the other player to use the naive (GDA) strategy, or as decreasing the other players influence (by decreasing their gradient).

3. The *consensus term* $\pm D_{xx}^2\nabla_x f$, $\mp D_{yy}^2\nabla_y f$ that determines whether the players prefer to decrease their gradient ($\pm\,=\,+$) or to increase it ($\pm\,=\,-$). The former corresponds the players seeking consensus, whereas the latter can be seen as the opposite of consensus.
(It also corresponds to an approximate Newton's method. [4])

4. The *equilibrium term* $(\mathrm{Id}+\eta^2 D_{xy}^2 D_{yx}^2 f)^{-1}$, $(\mathrm{Id}+\eta^2 D_{yx}^2 D_{xy}^2 f)^{-1}$, which arises from the players solving for the Nash equilibrium. This term lets each player prefer strategies that are less vulnerable to the actions of the other player.

Each of these is responsible for a different feature of the corresponding algorithm, which we can illustrate by applying the algorithms to three prototypical test cases considered in previous works.

• We first consider the bilinear problem $f(x,y)=\alpha xy$ (see Figure 2). It is well known that GDA will fail on this problem, for any value of $\eta$. For $\alpha=1.0$, all the other methods converge exponentially towards the equilibrium, with ConOpt and SGA converging at a faster rate due to the stronger gradient correction ($\gamma>\eta$). If we choose $\alpha=3.0$, OGDA, ConOpt, and SGA fail. The former diverges, while the latter two begin to oscillate widely. If we choose $\alpha=6.0$, all methods but CGD diverge.

• In order to explore the effect of the consensus Term 3, we now consider the convex-concave problem $f(x,y)=\alpha(x^2-y^2)$ (see Figure 3). For $\alpha=1.0$, all algorithms converge at an exponential rate, with ConOpt converging the fastest, and OGDA the slowest. The consensus promoting term of ConOpt accelerates convergence, while the competition promoting term of OGDA slows down the convergence. As we increase $\alpha$ to $\alpha=3.0$, the OGDA and ConOpt start failing (diverge), while the remaining algorithms still converge at an exponential rate. Upon increasing $\alpha$ further to $\alpha=6.0$, all algorithms diverge.

• We further investigate the effect of the consensus Term 3 by considering the concave-convex problem $f(x,y)=\alpha(-x^2+y^2)$ (see Figure 3). The critical point $(0,0)$ does not correspond to a Nash-equilibrium, since both players are playing their *worst possible strategy*. Thus it is highly undesirable for an algorithm to converge to this critical point. However for $\alpha=1.0$, ConOpt does converge to $(0,0)$ which provides an example of the consensus regularization introducing spurious solutions. The other algorithms, instead, diverge away towards infinity, as would be expected. In particular, we see that SGA is correcting the problematic behavior of ConOpt, while maintaining its better convergence rate in the first example. As we increase $\alpha$ to $\alpha\in\{3.0,6.0\}$, the radius of attraction of $(0,0)$ under ConOpt decreases and thus ConOpt diverges from the starting point $(0.5,0.5)$, as well.

The first experiment shows that the inclusion of the competitive Term 2 is enough to solve the cycling problem in the bilinear case. However, as discussed after Theorem 2.2, the convergence results of existing methods in the literature are not break down as the interactions between the players becomes too strong (for the given $\eta$). The first experiment illustrates that this is not just a lack of theory, but corresponds to an actual failure mode of the existing algorithms. The experimental results in Figure 5 further show that for input dimensions $m,n>1$, the advantages of CGD can not be recovered by simply changing the stepsize $\eta$ used by the other methods.
While introducing the competitive term is enough to fix the cycling behaviour of GDA, OGDA and ConOpt (for small enough $\eta$) add the additional consensus term to the update rule, with opposite signs.
In the second experiment (where convergence is desired), OGDA converges in a smaller parameter range than GDA and SGA, while only diverging slightly faster in the third experiment (where divergence is desired).
ConOpt, on the other hand, converges faster than GDA in the second experiment, for $\alpha=1.0$ however, it diverges faster for the remaining values of $\alpha$ and, what is more problematic, it converges to a spurious solution in the third experiment for $\alpha=1.0$.
Based on these findings, the consensus term with either sign does not seem to systematically improve the performance of the algorithm, which is why we suggest to only use the competitive term (that is, use LOLA/LCGD, or CGD, or SGA).

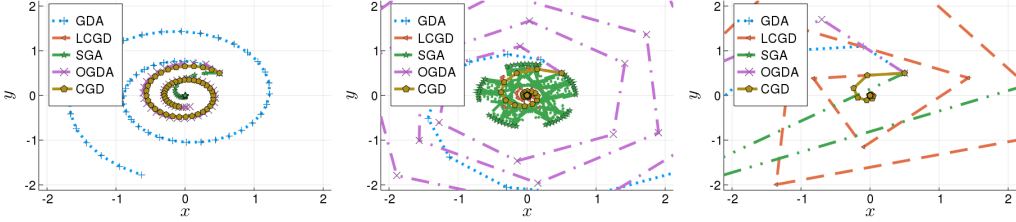

Figure 2: The first 50 iterations of GDA, LCGD, ConOpt, OGDA, and CGD with parameters $\eta = 0.2$ and $\gamma = 1.0$. The objective function is $f(x, y) = \alpha x^\top y$ for, from left to right, $\alpha \in \{1.0, 3.0, 6.0\}$. (Note that ConOpt and SGA coincide on a bilinear problem)

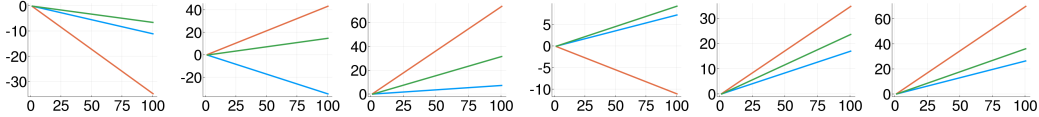

Figure 3: We measure the (non-)convergence to equilibrium in the separable convex-concave– ($f(x, y) = \alpha(x^2 - y^2)$, left three plots) and concave convex problem ($f(x, y) = \alpha(-x^2 + y^2)$, right three plots), for $\alpha \in \{1.0, 3.0, 6.0\}$. (Color coding given by GDA, SGA, LCGD, CGD, ConOpt, OGDA, the y-axis measures $\log_{10}(\|(x_k, y_k)\|)$ and the x-axis the number of iterations $k$. Note that convergence is desired for the first problem, while *divergence* is desired for the second problem.

## 4 Implementation and numerical results

We briefly discuss the implementation of CGD.

**Computing Hessian vector products:** First, our algorithm requires products of the mixed Hessian $v \mapsto D_{xy} f v, v \mapsto D_{yx} g v$, which we want to compute using automatic differentiation. As was already observed by Pearlmutter (1994), Hessian vector products can be computed at minimal overhead over the cost of computing gradients, by combining forward– and reverse mode automatic differentiation. To this end, a function $x \mapsto \nabla_y f(x, y)$ is defined using reverse mode automatic differentiation. The Hessian vector product can then be evaluated as $D_{xy}^2 f v = \frac{\partial}{\partial h} \nabla_y f(x + hv, y)\big|_{h=0}$, using forward mode automatic differentiation. Many AD frameworks, like Autograd (https://github.com/HIPS/autograd) and ForwardDiff(https://github.com/JuliaDiff/ForwardDiff.jl, (Revels et al., 2016)) together with ReverseDiff(https://github.com/JuliaDiff/ReverseDiff.jl) support this procedure. In settings where we are only given access to gradient evaluations but cannot use automatic differentiation to compute Hessian vector products, we can instead approximate them using finite differences.

**Matrix inversion for the equilibrium term**: Similar to a *truncated Newton's method* (Nocedal and Wright, 2006), we propose to use iterative methods to approximate the inverse-matrix vector products arising in the equilibrium term 4. We will focus on zero-sum games, where the matrix is always symmetric positive definite, making the conjugate gradient (CG) algorithm the method of choice. For nonzero sum games we recommend using the GMRES or BCGSTAB (see for example Saad (2003) for details). We suggest terminating the iterative solver after a given relative decrease of the residual is achieved ($\|Mx - y\| \leq \epsilon \|x\|$ for a small parameter $\epsilon$, when solving the system $Mx = y$). In our experiments we choose $\epsilon = 10^{-6}$. Given the strategy $\Delta x$ of one player, $\Delta y$ is the optimal counter strategy which can be found without solving another system of equations. Thus, we recommend in each update to only solve for the strategy of one of the two players using Equation (3), and then use the optimal counter strategy for the other player. The computational cost can be further improved by using the last round's optimal strategy as a a warm start of the inner CG solve. An appealing feature of the above algorithm is that the number of iterations of CG adapts to the difficulty of solving the equilibrium term 4. If it is easy, we converge rapidly and CGD thus *gracefully reduces to LCGD*, at only a small overhead. If it is difficult, we might need many iterations, but correspondingly the problem would be very hard without the preconditioning provided by the equilibrium term.

**Experiment: Fitting a bimodal distribution:** We use a simple GAN to fit a Gaussian mixture model with two modes, in two dimensions (see supplement for details). We apply SGA, ConOpt ($\gamma = 1.0$), OGDA, and CGD for stepsize $\eta \in \{0.4, 0.1, 0.025, 0.005\}$ together with RMSProp ($\rho = 0.9$). In

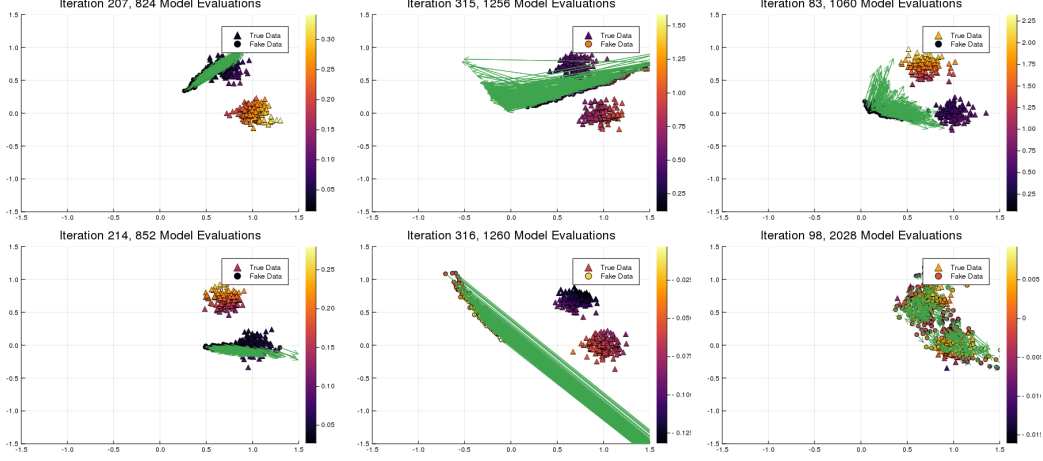

Figure 4: For all methods, initially the players cycle between the two modes (first column). For all methods but CGD, the dynamics eventually become unstable (middle column). Under CGD, the mass eventually distributes evenly among the two modes (right column). (The arrows show the update of the generator and the colormap encodes the logit output by the discriminator.)

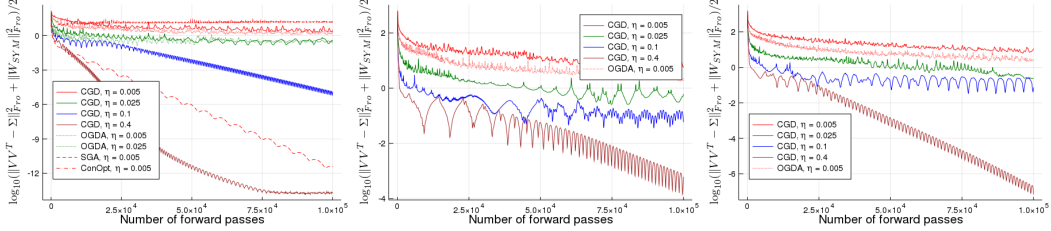

Figure 5: We plot the decay of the residual after a given number of model evaluations, for increasing problem sizes and $\eta \in \{0.005, 0.025, 0.1, 0.4\}$. Experiments that are not plotted diverged.

each case, CGD produces an reasonable approximation of the input distribution without any mode collapse. In contrast, all other methods diverge after some initial cycling behaviour! Reducing the steplength to $\eta = 0.001$, did not seem to help, either. While we do not claim that the other methods can not be made work with proper hyperparameter tuning, this result substantiates our claim that CGD is significantly more robust than existing methods for competitive optimization. For more details and visualizations of the whole trajectories, consult the supplementary material.

**Experiment: Estimating a covariance matrix:** To show that CGD is also competitive in terms of computational complexity we consider the noiseless case of the covariance estimation example used by Daskalakis et al. (2017)[Appendix C], We study the tradeoff between the number of evaluations of the forward model (thus accounting for the inner loop of CGD) and the residual and observe that for comparable stepsize, the convergence rate of CGD is similar to the other methods. However, due to CGD being convergent for larger stepsize it can beat the other methods by more than a factor two (see supplement for details).

## 5 Conclusion and outlook

We propose a novel and natural generalization of gradient descent to competitive optimization. Besides its attractive game-theoretic interpretation, the algorithm shows improved robustness properties compared to the existing methods, which we study using a combination of theoretical analysis and computational experiments. We see four particularly interesting directions for future work. First, we would like to further study the practical implementation and performance of CGD, developing it to become a useful tool for practitioners to solve competitive optimization problems. Second, we would like to study extensions of CGD to the setting of more than two players. As hinted in Section 2, a natural candidate would be to simply consider multilinear quadratically regularized local models, but

the practical implementation and evaluation of this idea is still open. Third, we believe that second order methods can be obtained from biquadratic approximations with cubic regularization, thus extending the cubically regularized Newton's method of Nesterov and Polyak (2006) to competitive optimization. Fourth, a convergence proof in the nonconvex case analogue to Lee et al. (2016) is still out of reach in the competitive setting. A major obstacle to this end is the identification of a suitable measure of progress (which is given by the function value in the setting in the single agent setting), since norms of gradients can not be expected to decay monotonously for competitive dynamics in non-convex-concave games.

**Acknowledgments**

A. Anandkumar is supported in part by Bren endowed chair, Darpa PAI, Raytheon, and Microsoft, Google and Adobe faculty fellowships. F. Schäfer gratefully acknowledges support by the Air Force Office of Scientific Research under award number FA9550-18-1-0271 (Games for Computation and Learning) and by Amazon AWS under the Caltech Amazon Fellows program. We thank the reviewers for their constructive feedback, which has helped us improve the paper.

## Footnotes

[1]Here and in the following, unless otherwise mentioned, all derivatives are evaluated in the point $(x_k, y_k)$

[2] We could alternatively use the penalty $(x^\top x + y^\top y)/(2\eta)$ for both players, without changing the solution.

[3] We note that the matrix inverses exist for all but one value of $\eta$, and for all $\eta$ in the case of a zero sum game.

[4]Applying a damped and regularized Newton's method to the optimization problem of Player 1 would amount to choosing $x_{k+1}=x_k-\eta(\mathrm{Id}+\eta D_{xx}^2)^{-1}f\nabla_x f\approx x_k-\eta(\nabla_x f-\eta D_{xx}^2 f\nabla_x f)$, for $\|\eta D_{xx}^2 f\|\ll 1$.

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
