[Supplementary Material · supplement_final.pdf]

# Supplement: Competitive Gradient Descent

## Abstract

1      This is the supplement to the paper "Competitive Gradient Descent"

## 1   Proofs of convergence

3  *Proof of Theorem 2.3.* To shorten the expressions below, we set $a := \nabla_x f(x_k)$, $b := \nabla_y f(x_k, y_k)$,
4  $H_{xx} := D^2_{xx} f(x_k, y_k)$, $H_{yy} := D^2_{yy} f(x_k, y_k)$, $N := D^2_{xy} f(x_k, y_k)$, $\tilde{N} := \eta N$, $\tilde{M} := \tilde{N}^\top \tilde{N}$, and
5  $\bar{M} := \tilde{N} \tilde{N}^\top$. Letting $(x, y)$ be the update step of CGD and using Taylor expansion, we obtain

$$
\begin{aligned}
&(\nabla_x f(x + x_k, y + y_k))^2 + (\nabla_y f(x + x_k, y + y_k))^2 - \|a\|^2 - \|b\|^2 \\
\leq\, & 2x^\top H_{xx} a + 2x^\top N b + 2a^\top N y + 2b^\top H_{yy} y \\
& + 4L(\|x\|^2 + \|y\|^2)(\|a\| + \|b\|) \\
=\, & + 2\eta \left( -a^\top - b^\top \tilde{N}^\top \right) \left( \mathrm{Id} + \bar{M} \right)^{-1} H_{xx} a \\
& + 2x^\top N b + 2a^\top N y \\
& + 2\eta b^\top H_{yy} \left( \mathrm{Id} + \tilde{M} \right)^{-1} \left( b - \tilde{N}^\top a \right) \\
& + 4L(\|x\|^2 + \|y\|^2)(\|a\| + \|b\|) = \dots,
\end{aligned}
$$

6  By expanding zero to $\pm 2\eta b^\top \tilde{N}^\top \left( \mathrm{Id} + \bar{M} \right)^{-1} H_{xx} a$ and $\pm 2\eta b^\top H_{yy} \left( \mathrm{Id} + \tilde{M} \right)^{-1} \tilde{N}^\top a$, we obtain

$$
\begin{aligned}
\dots =\, & -2\eta a^\top H_{xx} a + 2\eta a^\top \bar{M} \left( \mathrm{Id} + \bar{M} \right)^{-1} H_{xx} a \\
& - 2\eta b^\top \tilde{N}^\top \left( \mathrm{Id} + \bar{M} \right)^{-1} H_{xx} a \\
& + 2x^\top N b + 2a^\top N y \\
& + 2\eta b^\top H_{yy} b + b^\top H_{yy} \left( \mathrm{Id} + \tilde{M} \right)^{-1} \tilde{M} b \\
& - 2\eta b^\top H_{yy} \left( \mathrm{Id} + \tilde{M} \right)^{-1} \tilde{N}^\top a \\
& + 4L(\|x\|^2 + \|y\|^2)(\|a\| + \|b\|) = \dots.
\end{aligned}
$$

7  We now plug the update rule of CGD into $x$ and $y$ and observe that $\tilde{N}^\top (\mathrm{Id} + \bar{M})^{-1} =$
8  $(\mathrm{Id} + \tilde{M})^{-1} \tilde{N}^\top$ to obtain

$$
2x^\top N b + 2a^\top N y = -2a^\top \left( \mathrm{Id} + \bar{M} \right)^{-1} \bar{M} a - 2b^\top \left( \mathrm{Id} + \tilde{M} \right)^{-1} \tilde{M} b.
$$

By plugging this into our main computation, we obtain

$$
\begin{aligned}
\ldots = & -2\eta a^\top H_{xx} a + 2\eta a^\top \bar{M} \left(\mathrm{Id} + \bar{M}\right)^{-1} H_{xx} a \\
& - 2\eta b^\top \tilde{N}^\top \left(\mathrm{Id} + \bar{M}\right)^{-1} H_{xx} a \\
& - 2a^\top \left(\mathrm{Id} + \bar{M}\right)^{-1} \bar{M} a - 2b^\top \left(\mathrm{Id} + \tilde{M}\right)^{-1} \tilde{M} b \\
& + 2\eta b^\top H_{yy} b - 2\eta b^\top H_{yy} \left(\mathrm{Id} + \tilde{M}\right)^{-1} \tilde{M} b \\
& - 2\eta b^\top H_{yy} \left(\mathrm{Id} + \tilde{M}\right)^{-1} \tilde{N}^\top a \\
& + 4L(\|x\|^2 + \|y\|^2)(\|a\| + \|b\|) \leq \ldots.
\end{aligned}
$$

By positivity of squares, we have

$$
2\eta a^\top \bar{M} \left(\mathrm{Id} + \bar{M}\right)^{-1} H_{xx} a \leq a^\top \left(\bar{M} \left(\mathrm{Id} + \bar{M}\right)^{-1}\right)^2 a + a^\top \left(\eta H_{xx}\right)^2 a
$$

$$
-2\eta b^\top H_{yy} \left(\mathrm{Id} + \tilde{M}\right)^{-1} \tilde{M} b \leq b^\top \left(\tilde{M} \left(\mathrm{Id} + \tilde{M}\right)^{-1}\right)^2 b + b^\top \left(\eta H_{yy}\right)^2 b.
$$

For $\lambda \in [-1, 1]$ we have $-2\lambda + \lambda^2 = 2\lambda(1 - \lambda/2) \leq -h_\pm(\lambda))$ from which we deduce the result. $\qquad\square$

Theorem 2.4 follows from Theorem 2.3 by relatively standard arguments:

*Proof of Theorem 2.4.* Since $\nabla_x f(x^*, y^*), \nabla_x f(x^*, y^*) = 0$ and the gradient and Hessian of $f$ are continuous, there exists a neighbourhood $\mathcal{V}$ of $(x^*, y^*)$ such that for all possible starting points $(x_1, y_1) \in \mathcal{V}$, we have $\|(\nabla_x f(x_2, y_2), \nabla_y f(x_2, y_2)\| \leq (1 - \lambda_{\min}/4)\|(\nabla_x f(x_1, y_1), \nabla_y f(x_1, y_1)\|$. Then, by convergence of the geometric series there exists a closed neighbourhood $\mathcal{U} \subset \mathcal{V}$ of $(x^*, y^*)$, such that for $(x_0, y_0) \in \mathcal{U}$ we have $(x_k, y_k) \in \mathcal{V}, \forall k \in \mathbb{N}$ and thus $(x_k, y_k)$ converges at an exponential rate to a point in $\mathcal{U}$. $\qquad\square$

# 2 Details regarding the experiments

## 2.1 Experiment: Estimating a covariance matrix

We consider the problem $-g(V, W) = f(W, V) = \sum_{ijk} W_{ij} \left(\hat{\Sigma}_{ij} - (V\hat{\Sigma}V^\top)_{i,j}\right)$, where the $\hat{\Sigma}$ are empirical covariance matrices obtained from samples distributed according to $\mathcal{N}(0, \Sigma)$. For our experiments, the matrix $\Sigma$ is created as $\Sigma = UU^T$, where the entries of $U \in \mathbb{R}^{d \times d}$ are distributed i.i.d. standard Gaussian. We consider the algorithms OGDA, SGA, ConOpt, and CGD, with $\gamma = 1.0$, $\epsilon = 10^{-6}$ and let the stepsizes range over $\eta \in \{0.005, 0.025, 0.1, 0.4\}$. We begin with the deterministic case $\hat{\Sigma} = \Sigma$, corresponding to the limit of large sample size. We let $d \in \{20, 40, 60\}$ and evaluate the algorithms according to the trade-off between the number of forward evaluations and the corresponding reduction of the residual $\|W + W^\top\|_{\mathrm{FRO}}/2 + \|UU^\top - VV^\top\|_{\mathrm{FRO}}$, starting with a random initial guess (the same for all algorithms) obtained as $W_1 = \delta W$, $V_1 = U + \delta V$, where the entries of $\delta W, \delta V$ are i.i.d uniformly distributed in $[-0.5, 0.5]$. We count the number of "forward passes" per outer iteration as follows.

- OGDA: 2
- SGA: 4
- ConOpt: 6
- CGD: 4 + 2 ∗ number of CG iterations

The results are summarized in Figure 1. We see consistently that for the same stepsize, CGD has convergence rate comparable to that of OGDA. However, as we increase the stepsize the other methods start diverging, thus allowing CGD to achieve significantly better convergence rates by using larger stepsizes. For larger dimensions ($d \in \{40, 60\}$) OGDA, SGA, and ConOpt become

Figure 1: The decay of the residual as a function of the number of forward iterations ($d = 20, 40, 60$, from top to bottom). **Note that missing combinations of algorithms and stepsizes correspond to divergent experiments**. While the exact behavior of the different methods is subject to some stochasticity, results as above were typical during our experiments.

even more unstable such that OGDA with the smallest stepsize is the only other method that still converges, although at a much slower rate than CGD with larger stepsizes. We now consider the stochastic setting, where at each iteration a new $\hat{\Sigma}$ is obtained as the empirical covariance matrix of $N$ samples of $\mathcal{N}(0, \Sigma)$, for $N \in \{100, 1000, 10000\}$. In this setting, the stochastic noise very quickly dominates the error, preventing CGD from achieving significantly better approximations than the other algorithms, while other algorihtms decrease the error more rapidly, initially. It might be possible to improve the performance of our algorithm by lowering the accuracy of the inner linea system solve, following the intuition that in a noisy environment, a very accurate solve is not worth the cost. However, even without tweaking $\epsilon$ it is noticable than the trajectories of CGD are less noisy than those of the other algorithms, and it is furthermore the only algorithm that does not diverge for any of the stepsizes. It is interesting to note that the trajectories of CGD are consistently more regular than those of the other algorithms, for comparable stepsizes.

## 2.2 Experiment: Fitting a bimodal distribution

We use a GAN to fit a Gaussian mixture of two Gaussian random variables with means $\mu_1 = (0, 1)^\top$ and $\mu_2 = (2^{-1/2}, 2^{-1/2})^\top$, and standard deviation $\sigma = 0.1$ Generator and discriminator are given by dense neural nets with four hidden layers of $128$ units each that are initialized as orthonormal matrices, and ReLU as nonlinearities after each hidden layer. The generator uses $512$-variate standard Gaussian noise as input, and both networks use a linear projection as their final layer. At each step, the discriminator is shown $256$ real, and $256$ fake examples. We interpret the output of the discriminator as a logit and use sigmoidal crossentropy as a loss function. We tried stepsizes $\eta \in \{0.4, 0.1, 0.025, 0.005\}$ together with RMSProp ($\rho = 0.9$) and applied SGA, ConOpt ($\gamma = 1.0$), OGDA, and CGD. Note that the RMSProp version of CGD with diagonal scaling given by the matrices $S_x$, $S_y$ is obtained by replacing the quadratic penalties $x^\top x/(2\eta)$ and $y^\top y/(2\eta)$ in the local game by $x^\top S_x^{-1} x/(2\eta)$ and $y^\top S_x^{-1} y/(2\eta)$, and carrying out the remaining derivation as before. This also allows to apply other adaptive methods like ADAM. On all methods, the generator and discriminator are initially chasing each other across the strategy space, producing the typical cycling pattern. When using SGA, ConOpt, or OGDA, however, eventually the algorithm diverges with the generator either mapping all the mass far away from the mode, or collapsing the generating map to become zero. Therefore, we also tried decreasing the stepsize to $0.001$, which however did not prevent the divergence. For CGD, after some initial cycles the Generator starts splitting the mass and distributes is roughly evenly among the two modes. During our experiments, this configuration appeared to be robust. In the supplement, we have included a number of visualizations of the games trajectories for a variety of stepsizes and algorithms. Here, for example the folder with the name `two_mode_conOpt_25` contains the experiment with ConOpt and stepsize $\eta = 25 * 0.001 = 0.025$.

Figure 2: The decay of the residual as a function of the number of forward iterations in the stochastic case with $d = 20$ and batch sizes of $100, 1000, 10000$, from top to bottom).