[Reviews · NeurIPS 2019]

Reviewer 1



This paper deals with the computation of Nash Equilibria in competitive two-player games where the $x$ player is minimizing a function $f(x,y)$ and the $y$ player is minimizing a function $g(x,y)$. Such problems arise in a wide variety of domains, notably in training GANs, and there has been much recent interest in developing algorithms for solving such problems. Gradient Descent Ascent (GDA) is a natural candidate algorithm for finding Nash Equilibrium, but it will provably oscillate or diverge even in simple settings. As such, many recent works have modified GDA or proposed different algorithms or schemes to guarantee convergence. This paper proposes a new algorithm called Competitive Gradient Descent (CGD), which updates each player's iterates by adding the Nash Equilibrium of a regularized bilinear approximation of the game at the current iterates. CGD requires Hessian-vector products, making it a second-order algorithm. The authors prove that their algorithm is locally stable around critical points for zero-sum games. They discuss how their algorithm relates to other recent algorithms for finding Nash Equilibria, and they also provide a number of experimental comparisons. Unlike other algorithms, CGD does not need to reduce its step-size parameter $\eta$ when the interaction'' between the players is large, as signified by the magnitude of $D^2_{xy} f$ being large. On the positive side, this paper proposes a novel algorithm that seems reasonably well-motivated. Moreover, the paper gives a nice interpretation of how this algorithm compares to other existing algorithms. Finally, the experimental results are reasonably extensive and seem to indicate that CGD may have good empirical performance. On the negative side, the theoretical results are fairly weak, since they just prove stability near critical points. The authors emphasize how CGD improves when $D^2_{xy}$ is large, whereas other algorithms have poor performance for fixed step-size as $D^2_{xy}$ grows. However, this comparison isn't exactly fair since CGD uses second-order information to explicitly regularize step-sizes by the off-diagonal term, whereas OGDA or extragradient use only first-order information. Moreover, CGD still requires that the step-size is bounded by one over the max diagonal entry of the Hessian, so at least from the theory, CGD will still require step-size tuning. Finally, while it is true that Hessian-vector products are theoretically as fast as gradient calls for neural networks and explicitly defined functions, it is misleading to simply say Hessian vector products can be computed at minimal overhead over the cost of computing gradients'' because this is not true in general, for instance when one only has oracle access to the function and gradient. Overall, this paper is an accept. The algorithm is novel and well-motivated, and the authors do a good job of comparing it to existing results. While the theoretical results are minimal, the experimental results seem well-done and fairly thorough. The paper is clear. --- After Rebuttal --- I have read the other reviews and the rebuttal. I had no major concerns for the paper, so my rating is unchanged.

Reviewer 2



The writing of this paper is quite good, and the game-theoretic perspective is very much welcome. Theorem 2.2 is the main theoretical contribution of this work. The convergence analysis is nice, but it is not very surprising, nor is the proof technique itself very novel. The experiments are fair, but they are not the strength of this work. I have the following major concerns. 1. If the authors are willing to use a matrix inverse each iteration, why not consider the full 2nd order Taylor expansion and perform a (damped) Newton method? What benefit is there to only consider the quadratic interaction terms (bilinear terms) and not the individual quadratic terms? 2. Glaringly absent from this paper is the discussion of whether computing the matrix inverse (via iterative methods) is computationally worth it compared to existing first-order methods that do not require a matrix inverse. The computational experiments only demonstrate stability and rate of convergence as a function of the iteration count. For the x-axis, the wall-clock time should have been used instead of the iteration count. I imagine the method does not scale very well as the dimension of the variable gets large. Overall, I believe the theoretical contribution is nice but not very strong. As the experiments are minimal and do not address the concerns about the computational cost, they fail to provide a proof-of-concept of the practical value of the proposed method. Minor issues. line 181, the inline equation seems wrong. line 235 " are not break down" line 250, the idea of computing Hessian vector products with automatic differentiation is not an original idea of this work. Balduzzi et al, or whoever thought of it first should be credited. line 280 "an reasonable" -> "a reasonable" For GAN and covariance matrix estimation experiments, you should state in the text or within the figure captions indicating which figure corresponds to which experiment. Supplement. The theorem numberings are off by one. -------------------------------------------------------------------------------------------- Post rebuttal comments: The authors’ response has resolved my concerns regarding practical computational costs. I now see that the experiments do provide a proof of concept for the practical effectiveness of the proposed method. I still think my #1 concern of “why not Newton?” is still a significant weakness of this work, and I do not expect further discussion without any analysis or experiments to resolve this issue. Nevertheless, I think the presented results are interesting enough to be published.

Reviewer 3



A new algorithm, competitive gradient descent (CGD), is proposed to solve for the Nash equilibrium of two-player zero-sum games. It is derived from the closed form solution of a local bilinear approximation to the game at each learning step. Given assumptions such as bounds on the player Hessians, exponential convergence is proven. CGD is compared to several other algorithms in the literature both empirically and based on mathematical form. A link to levels of intentionality is also made using the Taylor series expansion of the inverse. Finally, experiments fitting a GAN to a Gaussian mixture (bi-modal) distribution show CGD converges faster than other methods. I believe the perspective of solving for the Nash equilibrium of a bilinear approximation is novel, however, an argument for the specific bilinear approximation proposed is lacking. Moreover, if the bilinear approximation were to retain the diagonal Hessian terms, e.g., 0.5*x^T (D^2_xx) x, the derived update becomes the standard regularized Newton method (based on a quick calculation): new preconditioning matrix from line 147 = [[I+eta*Dxx f, eta*Dxy f],[eta*Dyx g, I+eta*Dyy g]]^{-1} = (I + eta*Jacobian)^{-1}. This is regularized in the sense that identity better conditions the matrix for inversion. Therefore, the update presented on line 147 is nearly Newton's method applied to fixed point iteration, i.e., Delta = -J^{-1}F where J is the Jacobian. Theorem 2.2 requires eta*||D^2_xx f|| <= 1 -- does this condition arise because the diagonal terms in line 147 are replacing eta*(D^2_xx f) from Newton's method with identity? A discussion around these similarities would be helpful. It is known that Newton's method is attracted to critical points generally, not just minima (see "Identifying and attacking the saddle point problem in high-dimensional non-convex optimization"). By replacing the diagonal terms in line 147 with identity, CGD can avoid converging to some unstable fixed points -- does this prevent all failure modes? What are some of CGD's failure modes if any? Doesn't the requirement in Theorem 2.2 (eta*||D^2_xx f|| <= 1) suggest that a large enough step size, eta, may prevent convergence? The work is generally high quality and clear, and many of the perspectives are original. I think the community would benefit from some of the ideas in this paper. The paper would benefit from a discussion of the weaknesses of CGD. Main concerns: 1) CGD is very similar to Newton's method, but there is no rationale mentioned for dropping the diagonal terms of Jacobian. 2) No argument is presented for the specific bilinear form proposed as the fundamental local approximation to the game. 3) Complexity of inverting a matrix is dealt with using conjugate gradient but comparisons are done using "simple" GANs. Would this method scale well to higher dimensions? Do you believe there is a threshold at which the matrix inversion method becomes intractable? Have you tried CGD on more standard GAN tasks, e.g., MNIST, Celeb-A, CIFAR10? Do you expect problems at that scale? Also, SGA [Balduzzi et al, 2018] was independently derived in "Global Convergence to the Equilibrium of GANs using Variational Inequalities" [Gemp et al, 2018]. This work provides a comparison similar to Figure 1 in this paper. It should probably be cited along with Balduzzi '18 and Letcher '19.

[Author Response · NeurIPS 2019]

We thank all three reviewers for their thorough reviews and constructive feedback.

**Why not Newton?:**
As remarked by reviewers #6 and #7, retaining the diagonal blocks of the Hessian in the approximation results in a
regularized and damped Newton method $(\Delta x, \Delta y) = (\mathrm{Id} + \eta J)^{-1}(\nabla_x f, \nabla_y g)$, where $J$ is the Jacobian of the vector
field $(\nabla_x f, \nabla_y g)$. We agree that the present version of the paper lacks a thorough discussion of the reasons for ignoring
the diagonal parts of the Hessian and will therefore add the following three reasons.
**Blow-up of condition number:** Including the diagonal blocks of the Hessian in the non-convex-concave setting can
make the matrix inverse arbitrarily ill-conditioned as $\|\eta D_{xx}^2 f\|$ or $\|\eta D_{yy}^2 f\|$ approach or exceed 1, greatly increasing
the cost of the linear system solve. In contrast, for zero-sum games the condition number of the matrix inverse in CGD
is always bounded from above by $(1 + \eta^2 \|D_{xy}^2 f\|^2)$.
**Irrational updates:** For $\eta \|D_{xx}^2 f\|$ or $\eta \|D_{yy}^2 f\|$ bigger than 1 and $f$ non-convex-concave, the regularized Newton
update can loose its game-theoretic interpretation as a local strategic equilibrium, allowing for convergence to highly
non-optimal critical points. While we leave the full characterization of the attractors of CGD for future work, we expect
them to always be game-theoretically meaningful since the updates of CGD arise as local Nash equilibria.
**Lack of regularity:** For the diagonal blocks of the Hessian to be useful in optimization, we need to make additional
assumptions on the regularity of the loss function, for example by bounding the Lipschitz constants of $D_{xx}^2 f, D_{yy}^2 f$.
Otherwise, including additional second order information can make the results worse. Consider for instance, minimizing
$x \mapsto x^2 + \epsilon^{3/2} \sin(x/\epsilon)$ for $\epsilon \ll 1$. Many minimax problems, for example GANs, have the form $f(x, y) =$
$\Phi(\mathcal{G}(x), \mathcal{D}(y))$ where $\Phi$ is *smooth* and *simple* but $\mathcal{G}$ and $\mathcal{D}$ might only have first order regularity. In this setting, the
bilinear approximation has the advantage of fully exploiting first order information of $\mathcal{G}, \mathcal{D}$, without assuming them to
have higher degrees of regularity. This is because the bilinear approximation of $f$ then contains only the first derivatives
of $\mathcal{G}$ and $\mathcal{D}$, while the quadratic approximation contains second derivatives $D_{xx}^2 \mathcal{G}$ and $D_{yy}^2 \mathcal{D}$, and therefore needs
stronger regularity assumptions on $\mathcal{G}$ and $\mathcal{D}$ to be effective.

**Reviewer #3:**
*"...convergence rate results for CGD..."*: Under lower (upper) bounds on $D_{xx} f$ ($D_{yy} f$), global exponential conver-
gence can be derived from Theorem 2.2, and we are happy to include this result with the revisions. A special case of this
is strong convex-concavity. We are not aware of existing work on minimaximization that provides global convergence
proofs without either convex-concavity/monotonicity, or strong additional assumptions.
*"...CGD still requires that the step-size is bounded by one over the max diagonal entry of the Hessian..."*: Correct!
This is the analogous requirement to applying gradient descent to the single player game (keeping the other player
fixed). For problems like GANs the problem of optimizing one player while keeping the other player fixed can be
solved reliably via gradient descent while the two-player game becomes unstable under alternating gradient descent.
The purpose of CGD is to solve two-player games with similar step sizes and stability properties as when using gradient
descent to optimize one player while the other player is kept fix.

**Reviewer #6:**
**Concern 1:** *Why not use full second order?* See first paragraph.
**Concern 2:** *Complexity of matrix inverse?* We are sorry for the misunderstanding and will try to make line 289 more
precise: Figure 5 does **not** show the convergence as a function of the iteration count, but as a function of the number
of gradient evaluations and Hessian-vector products. Thus, a single step of CGD that needs $k$ iterations of conjugate
gradient to solve the linear system in the update rule will amount to an $x$-value of $(4 + 2k)$, while a single step of
optimistic gradient descent ascent (OGDA) corresponds to an $x$-value of 2 in the plot. Thus, this measure of cost fairly
accounts for the complexity of the matrix inverse in CGD. We find that the convergence rate of CGD is competitive
throughout a wide range of step sizes and we explain this fact with the *graceful reduction to linearized CGD* described
in Line 274: If the gain from the matrix inverse is small, the matrix will be well-conditioned and thus easy to invert.
See also our answer to Reviewer #7.

**Reviewer #7:**
**Concerns 1 & 2:** *Why drop diagonal blocks of Hessian? Why use bilinear approximation?* See first paragraph.
**Concern 3:** *Is CGD scalable?* Since mixed mode automatic differentiation allows to compute Hessian vector products
with minimal overhead compared to gradient computations using reverse mode automatic differentiation, we see no
reason why CGD should be restricted to small problems. While larger problems will tend to require more iterations of
conjugate gradient, they also tend to negatively affect the existing methods, with the experiments in Figure 5 suggesting
that the advantage of CGD over existing methods *increases* as the problems size increases. We are presently working
on an implementation of CGD using JAX (which provides GPU accelerated mixed mode automatic differentiation) to
then apply to large GAN problems.

[Meta-Review · NeurIPS 2019]

Overall, the reviewers appreciated the novelty and simplicity of the algorithm. The paper is also well written and the methods are well motivated. In terms of content for revision: The lack of discussion around Newton-style approximation and dropping of diagonal terms in the Jacobian was raised again the discussions (see updated reviews). Additionally, we also sought an expert reviewer during discussions. The additional reviewer included below: ---- Additional review: The paper proposes an interesting new method for solving games Min_x F(x,y) Min_y G(x,y) which is, in general, different from running online gradient descent on x and y in parallel, or other existing methods. The idea is to do a bilinear approximation of F and G and solve for the equilibrium of the resulting game (with regularization) then add the equilibrium (x^*,y^*) of this game to the current iterate (x_t,y_t) to get the next iterate. It’s an interesting and natural idea. The abstract oversells the result in the following ways: - they claim their method avoids oscillatory behavior - their method does not need to adapt the step size However: - they only show *local convergence results* and in particular their method may as well oscillate if initialized away from the equilibrium (in contrast to other methods such as optimistic gradient descent or extragradient which have been shown to have last iterate convergence for convex-concave problems) - they need to choose their eta appropriately for this to hold as also have to do other methods Their empirical findings for bilinear games alpha*x*y with scalar x and y show better stability than other methods for a fixed learning rate as alpha is tuned higher. I would expect that as alpha is tuned higher their method eventually diverges. Overall, I think it is an interesting proposal for a method to solve bilinear games, but there is a lot of overselling in the paper around non-oscillatory behavior and around the lack of need to do hyper-parameter tuning. I would measure those claims and also try to prove global convergence results for convex-concave problems. The experiments are encouraging.